# Oral Chronic Hyperplastic Candidiasis and Its Potential Risk of Malignant Transformation: A Systematic Review and Prevalence Meta-Analysis

**DOI:** 10.3390/jof8101093

**Published:** 2022-10-17

**Authors:** Alejandro I. Lorenzo-Pouso, Alba Pérez-Jardón, Vito Carlo Alberto Caponio, Francesca Spirito, Cintia M. Chamorro-Petronacci, Óscar Álvarez-Calderón-Iglesias, Pilar Gándara-Vila, Lorenzo Lo Muzio, Mario Pérez-Sayáns

**Affiliations:** 1Oral Medicine, Oral Surgery and Implantology Unit (MedOralRes Group), Faculty of Medicine and Dentistry, University of Santiago de Compostela, 15782 A Coruña, Spain; 2Health Research Institute of Santiago de Compostela (IDIS), 15706 Santiago de Compostela, Spain; 3Department of Clinical and Experimental Medicine, University of Foggia, 71122 Foggia, Italy; 4Research, Health and Podiatry Group, Department of Health Sciences, Faculty of Nursing and Podiatry, University of A Coruña, 15008 A Coruña, Spain; 5HM Hospitals Research Foundation, 28015 Madrid, Spain

**Keywords:** malignant transformation, meta-analysis, mouth neoplasm, *Candida*, chronic hyperplastic candidiasis

## Abstract

Chronic hyperplastic candidiasis (CHC) is a prototypical oral lesion caused by chronic *Candida* infection. A major controversy surrounding CHC is whether this oral lesion owns malignant transformation (MT) potential. The aim of the present study was to evaluate current evidence on the MT of CHC and to determine the variables which have the greatest influence on cancer development. Bibliographical searches included PubMed, Embase, Web of Science, Scopus and LILACS. The cohort studies and case series used to investigate the MT of CHC were deemed suitable for inclusion. The quality of the enrolled studies was measured by the Joanna Briggs Institute scale. Moreover, we undertook subgroup analyses, assessed small study effects, and conducted sensitivity analyses. From 338 studies, nine were finally included for qualitative/quantitative analysis. The overall MT rate for CHC across all studies was 12.1% (95% confidential interval, 4.1–19.8%). Subgroup analysis showed that the MT rate increased when pooled analysis was restricted to poor quality studies. It remains complex to affirm whether CHC is an individual and oral, potentially malignant disorder according to the retrieved evidence. Prospective cohort studies to define the natural history of CHC and a consensus statement to clarify a proper set of diagnostic criteria are strongly needed. PROSPERO ID: CRD42022319572.

## 1. Introduction

*Candida* species (spp.) are a relevant group of eukaryotic diploid sexual yeasts involved in the relevant burden of human pathologies. These yeasts are frequently encountered in the oral microbiota of healthy individuals, with an incidence between 35 and 80%, depending on the population sampled [1]. While *Candida* spp. are normally harmless commensals, they can sometimes switch to pathogenic behavior. As a result, they contribute to the appearance of a group of pathologies commonly referred to as oral candidosis or candidiasis (OCs) [2]. OCs pathogenesis is driven by the elevated production of virulence determinants and the subsequent impairment of the host immune responses [3].

Chronic hyperplastic candidosis or candidiasis (CHC) is a subtype of OCs and is considered a clinicopathological term, housing an intraoral white lesion generated by a persisting fungal infection, commonly *Candida albicans* [4]. Two clinical presentations have been described for this entity: a homogeneous form characterized by an isolated, adherent, and thick white plaque and a nodular/speckled form arising with multiple white nodules on an erythematous background. CHC commonly appears in the post-commissural buccal mucosa, the upper surface of the tongue, and the velum of denture users [1,3,4]. Although reported CHC incidence is low among OCs (about 1.61%) [5], its differential diagnosis is challenging since its clinical manifestations resemble the presentation of more common diseases, such as oral lichen planus and erythro-leukoplakia [6]. Another issue stands in the nomenclature. Since the first description of this entity by Cawson and Lehner back in 1968, the term has been considered a synonym of “candida leukoplakia” [7]; however, the role of this yeast infection in oral leukoplakia (OL) is still controversial [4]. At last, more importantly, CHC malignant transformation is still debated.

In terms of prevention strategies for CHC, there is poor quality of evidence. Patients should refrain from smoking or chewing tobacco. Some authors algo suggest the relevance of requesting patients to not wear their dentures at night and improve their oral hygiene. Some systemic problems can also contribute to the development of this type of candidiasis, such as anaemia, avitaminosis, and deficiencies in the immune system [2]. Various modalities of treatment for CHC have been used such as medical management in the form of antifungal therapy or the topical application of retinoids, bleomycin, or beta carotenes. On the other side, surgical methods have been used including cold-knife surgery, laser therapy, or cryosurgery [4].

For these reasons, CHC differential diagnosis is complex and time-consuming, requiring histopathology to exclude cellular atypia and combining the use of laboratory test cultures for fungal identification [8]. Indeed, over 15% of CHCs present histopathological features of epithelial dysplasia (ED) and higher malignant development rates than classical forms of OL lacking a *Candida* spp. Infection [3,9]. As is well known, OL is the most frequently encountered and emblematic oral potentially malignant disorder (OPMD). It is classically defined as “a white plaque with a questionable risk of cancer that can only be diagnosed once other specific conditions have been ruled out” [4]. *Candida* spp., particularly *albicans*, are capable of promoting oncogenesis by mechanisms such as carcinogenic by-products, molecular mimicry, or triggering the inflammation cascade [10]. As it is well known, the histopathological findings of ED in the first biopsy of any oral lesion are a strong predictor of future malignant transformation (MT) [11]. However, features of ED are not exclusive to premalignant lesions, and such histologic patterns may be reactive rather than pre-neoplastic [12]. The WHO Collaborating Centre for Oral Cancer noted that the OL definition is ad excludendum, which rules out specific causes such as *Candida* spp. infection. In this scenario, CHC per se is no longer defined as an OPMD [13].

Prompted by the discussed literature, we designed this systematic review and meta-analysis in order to more precisely determine the proportion of MT in the natural history of this particular affection.

## 2. Materials and Methods

This systematic review protocol was designed and planned in advance by an author (A.I.L.P). The present review was elaborated in accordance with the preferred reporting method for systematic reviews (PRISMA) [14], as prospectively registered at the PROSPERO database (ID: CRD42022319572).

### 2.1. Inclusion and Exclusion Criteria

The inclusion criteria were as follows: (a) Study design: observational studies with a prospective and retrospective design (i.e., cohort studies, case-control studies, or case series); (b) Participants: studies in humans of any sex or age diagnosis with a CHC; (c) Intervention: Proven histopathological diagnosis of *Candida* spp. or with cultural/or molecular approaches; (d) Outcome: incidence of malignant transformation in the sample studied.

Among exclusion criteria were: (a) Case reports, reviews, position papers, and author opinions; (b) Studies that did not accurately define CHC diagnosis both clinical and histopathological; (c) Studies that omitted *Candida* spp., detection methodology; (d) Studies that failed to provide the CHC MT rate or provided insufficient data to allow estimations; (e) reports including less than 3 CHC cases.

We included any relevant article according to the aforementioned criteria without restriction in terms of language or publication dates. In addition, we explored the references of included studies to complement our search strategy. Moreover, we established personal contact via e-mail with researchers to trace further publications or relevant data not included in their articles. This approach minimized the risk of introducing selection bias and improved the transparency, accuracy, and integrity of our methodology [14].

### 2.2. Search Strategy

Medline via PubMed, Embase, Web of Science, Scopus, and LILACS databases were screened for records published from inception to 22 June 2022 (upper limit). Our search strategy and syntax were conceptualized by combining thesaurus terms (e.g., MeSH or EMTREE) with free words. For Medline via PubMed, the following syntax was used both in MeSH and in the free terms: (“Candidiasis, Chronic Mucocutaneous” [MeSH] or “Chronic Hyperplastic Candidiasis” [All Fields] or “Candidal leukoplakia” [All Fields] or “Candida leukoplakia” [All Fields]) and (malign* or premalignant* or “potentially malignant disorder” or “precancer” or “cancer” [All Fields] or “Carcinoma, Squamous Cell” [MeSH] or “squamous cell carcinoma” [All Fields] or “oscc” [All Fields] or “transformation” [All Fields] or “risk” [All Fields] or “progression” [All Fields]).

The aforementioned algorithm was conveniently reformulated for each database, as can be seen in more detail in Table A1. Furthermore, to verify whether every relevant report dealing with the topic under study was retrieved, a complementary manual search with the terms: *Candida*, oral cancer, and malignant transformation in an unstructured form was executed.

### 2.3. Study Selection

In the first step, titles/abstracts of retrieved reports were screened on the basis of the inclusion/exclusion criteria using the Rayyan QCRI platform. In the second step, all selected studies that were considered proper according to these criteria underwent a full text assessment, and then a final decision was made. Excel spreadsheets were created to gather information regarding this decision-making process, to include/exclude each report. Cohen’s kappa coefficient (κ) was applied to quantify the reviewers’ intra-agreement at the full text screening process; this coefficient was calculated with an Epidat 4.2 statistical package (available at: https://www.sergas.es/Saude-publica/EPIDAT-4-2, accessed on 9 September 2022).

Study selection was performed by 2 independent operators (A.I.L.P. and M.P.S.) with expertise in oral medicine and pathology. In cases of disagreements among these reviewers, they were solved in joint meetings held with the participation of all authors. A third reviewer, blinded to the study hypothesis, performed an intra-agreement analysis by computing a value for κ (A.P.J.).

### 2.4. Data Extraction

Data were extracted with a specific form and improved by pilot-testing before its implementation. We collected information on author’s first name, publication date, country of origin, study design, population size, malignant transformation rates, recruitment/follow-up periods, method of *Candida* spp. detection, histopathological features, topography of lesions, sex, age, smoking, and alcohol intake among individuals. When data on an item were missing in the primary literature, attempts were made to contact the corresponding authors to incorporate lacking information.

Two reviewers (V.C.A.P. and M.P.S.) were responsible for the data extraction. This process was comprehensively reviewed later by a third participant (A.I.L.P.). In the event of disagreements, they were settled by careful discussion.

### 2.5. Critical Appraisal

The risk of bias (RoB) was established using a tool developed for systematic reviews and meta-analysis prevalence developed by the Joanna Briggs Institute (JBI), namely the JBI Prevalence Critical Appraisal Checklist [15]. This checklist assesses RoB studies by means of five domains in the form of closed questions: (i) Was the sample frame appropriate to address the target population? (ii) Were participants sampled appropriately? (iii) Was the sample size adequate for the study objective? (iv) Were participants and setting described properly? (v) Was the data analysis conducted with sufficient coverage? (vi) Were valid methods applied for the identification of the outcome? (vii) Was the outcome measured in a standard, reliable manner? (viii) Was there appropriate statistical analysis? (ix) Was the response rate adequate, or at least was the low response rate managed adequately? The studies were defined as a “high RoB” if 49% of items were reached with a “yes”; as “intermediate RoB” if a range between 50% and 69% of questions scored “yes”; and as “low RoB” if scores ≥ 70% received “yes”. In order to encounter a more detailed explanation regarding this scale, please see: https://jbi.global/sites/default/files/2019-05/JBI_Critical_Appraisal-Checklist_for_Prevalence_Studies2017_0.pdf, accessed on 9 September 2022.

### 2.6. Statistical Analysis

Study-specific proportions were computed with the portion of CHC cases reported with malignant transformation acting as numerators and the CHC cohort size as denominators. The 95% confidential intervals (CIs) were then estimated for retrieved studies with a simple asymptotic method with a continuity correction for the avoidance of mathematical aberrations or artifacts and to fuel the closeness of the coverage probability to its nominal value [16]. Later, study-specific log prevalence’s were weighted by the inverse of their variance to achieve pooled prevalence values with their corresponding 95% CIs. Meta analyses were performed with the DerSimonian and Laird method, which owed to logic dictated by some degree of between-study heterogeneity. Forest plots were generated to graphically display study-specific proportions as pooled proportions derived from the random-effects model. We also performed sensitivity analyses, evaluating the individual influence exerted by every study on the estimation of pooled rates; this approach was applied in order to check the reliability of combined results and to detect studies with aberrant results, bearing in mind the mathematical construct [17]. We also developed a mixed-effect meta-regression to check within-study variation and between-study variation to confront a specific covariate (year of publication) with the main outcome under study (i.e., MT rate).

Cochran’s Q test was implemented to assess between-study heterogeneity; *p* < 0.1 was considered a significant figure to assume heterogeneity. The I^2^ index was also used to quantify the percentage of heterogeneity. The thresholds established to define heterogeneity intensity were as follows: less than 25% no apparent, between 25% and 49% low, between 50% and 74% moderate, and 75% or greater high heterogeneity [18]. We later planned to explore the origin of heterogeneity across strata by restricting the analysis to subgroups defined by study-specific features, namely study origin and RoB assessment (high versus moderate and low RoB).

Publication bias or “small study effects” were evaluated visually using Begger’s funnel plot but also formally through the test described by Egger et al. [19]. The data analysis was performed using free R software (v.3.4.4; https://www.r-project.org, accessed on 9 September 2022, particularly using the Metafor package. A *p*-value less than 5% was considered statistically significant.

## 3. Results

### 3.1. Study Selection

Figure 1 displays the flow diagram of the electronic search, which resulted in 338 studies. After the screening of titles and abstracts, duplicates were removed, and 318 articles were excluded. After the screening of full texts, another 11 studies were excluded on the following basis. Eight articles were excluded because of their cross-sectional nature without any patient follow-up data. Two articles were excluded on the basis of being case reports, whilst one last study was discarded due to its representation of in vitro research results (i.e., off-topic). Nine articles were finally selected to enter the systematic review [6,7,20,21,22,23,24,25,26]. The reviewers’ intra-agreement was considered appropriate (κ = 0.83 [95% CI: 0.75–0.89].

### 3.2. Study Characteristics

Study characteristics in the qualitative synthetizes are shown in Table 1 and Table 2. The studies reported a total of 274 patients with CHC. The nine reports included in the present review were conducted in different countries across three continents: Europe [7,20,21,22,23,24], Asia [6,25], and North America [26]. Whilst these reports were published between 1966 and 2021, most studies were case series or retrospective cohort studies, and the target population was always clinic-based.

The main clinical and histopathological features of the 274 patients are displayed in Table 1 and Table 2. A total of seven studies stated the follow-up period of patients [6,7,21,22,24,25,26]. This time, they ranged between 2 and 16 years [22,24], with a mean of 8 years. All the retrieved studies reported cases of CHC with MT; this rate fluctuated broadly between 4.2% and 66.6% [6,23]. Among all the patients from the included studies, a total of 34 (12.4% of patients) progressed with a MT during the follow-up period.

Regarding the sex of CHC-affected patients, 68.7% were males, whilst 31.3% were females. Most of the patients were quinquagenarians and sexagenarians, with a mean age of 55.4. Lamentably, none of the primary literature reported relevant cofounders that related to MT properly, such as tobacco, alcohol, or betel consumption. The most frequent locations of CHC were buccal mucosa (58.9%), the tongue (15.7%), palate (13.7%), lip (7.8%) and gingiva (3.9%) (Table 2). Regarding ED assessment, different grading systems were applied owing to the studies. In this vein, we divided ED into mild, moderate, and severe, following the WHO proposal [27]. Only two of the included studies reported an ED grade or its absence [6,22], as exhaustively detailed in Table 2.

CHC ascertainment varied across the studies, although the gold standard was the periodic acid–Schiff stain (PAS) [6,7,21,22,24,25]. Moreover, three studies performed a culture to isolate *Candida* clinical species in blood agar, potato dextrose, and Sabouraud dextrose agar [20,23,26]. A single study was based on scraping coupled to the culture and further species identification [23].

### 3.3. Quality Scores and Data Synthesis

In summary, not all studies used for pooled analysis were conducted with the same vigor. One (11.1%) was classified as low [6], five (55.6%) as moderate [7,21,22,24,26] and three (33.3%) as a high RoB [20,23,25]. The domains showing a higher RoB potential were the condition diagnosis/measure (i.e., domain eight) and data analysis conducted with insufficient coverage of the primary data (i.e., domain five). However, hitherto studies also displayed a sub-optimal quality in some of the remaining domains, as extensively displayed in Figure 2.

Overall, the cumulative pooled prevalence of MT in CHC was 12.1% (95% CI, 4.1–19.8%; I^2^ = 78.27%, Q test *p* value = 0.002) (Figure 3A). A remarkable asymmetry could be seen in the forest plot, indicating a skewness to the right (Figure 3B). Egger’s regression test confirmed the existence of a publication bias (pEgger = 0.001). We stratified the subgroup analysis by either Asian or non-Asian, and the heterogeneity subsided for Asian studies (I^2^ = 31.03%), while the pooled estimate was also modified meaningfully [pooled proportion: 6.5% (95% CI, −1.5–14.5)]. On the other hand, in the non-Asian subgroup, heterogeneity was extremely large (I^2^ = 82.77). Stratification by a RoB assessment provided further insight; high-quality studies showed a pooled proportion of 9.8% (95 % CI: 2.0–17.8), while low-quality studies harbored a higher, albeit non-significant, proportion [pooled proportion = 34.6% (95% CI: −0.2–71.5) (Table 3).

In our sensitive analysis, the general results did not appreciably fluctuate after the sequential repetition of the pooled analysis with the omission of each individual study and this factor indicates that our estimations are not meaningfully influenced by a particular individual study. A single exception was noted as when Bánóczy et al. study [21] was omitted the pooled prevalence dropped to 6.6% (95% CI 1.1–12.1), which may explain part of the heterogeneity found (Table 4). The meta-regression, when executed, displayed a meaningful negative correlation between malignant transformation rates and the year of publication of the included reports (*p*-value = 0.001). A regression plot shows this feature in more detail [see Figure A1].

## 4. Discussion

Globally, the results of our systematic review displayed that one over ten patients diagnosed with chronic hyperplastic candidiasis will eventually undergo a malignant transformation. In this vein, the present study unraveled that CHC potential for malignant transformation is underestimated in the current scientific literature, with some authors considering CHC as a low-risk disorder [28,29]. Thus, it is key to perform an early diagnosis and control, as well as to monitor clinicopathological features that might be implicated in its malignancy [8].

CHC is a prototypical oral lesion caused by chronic *Candida* infection [30]. We consider that this infection is undervalued as a risk factor for squamous cell carcinoma, essentially because it is not registered in most epidemiological studies addressing the MT rate of apparently related conditions, such as canonical OL [31]. In the multifactorial context of oral carcinogenesis, assigning a causal role to *Candida* infection would dismiss the risk of the interaction between this risk factor with other well-known ones such as tobacco or alcohol [32]. In addition, the complex differential diagnosis and variety of nomenclatures for CHC implemented in the scientific literature may be responsible for a misdiagnosis of this disorder and, consequently, a misestimation of its malignant potential. However, we found a noticeable difference between the CHC and OL MT rates [31]. According to previous studies based on pooled analysis, the MT rate of OL ranges from 9.5% to 9.8% [33,34], whilst here we reported a rate of 12.1%. CHC may then accomplish the classical definition of a precancerous condition as ‘‘a morphologically alterated tissue in which cancer is more likely to occur than in apparently normal counterpart’’ [35]. However, *Candida* presence in OL lesions was considered as CHC in some of the retrieved scientific literature, and these conditions are considered histologically distinct, although they may frequently display a clinical overlap. This event entails a certain complexity for establishing a definite diagnostic threshold [4]. Overall, the present study demonstrates that our biologically plausible assumption seems to correlate epidemiologically by merging *Candida* infection as a remarkable risk factor for carcinogenesis. Indeed, nowadays, different theories exist on *Candida* infection mechanisms linked to oral carcinogenesis. *Candida* can produce nitrosamines such as N-nitroso-benzyl-methylamine and acetaldehyde, which are cancerogenic compounds [36,37]. Moreover, during infection, there is an increase in proinflammatory cytokines (interleukin (IL)-1α, IL-1β, IL-6, IL-8, IL-18, tumor necrosis factor (TNF)-α, IFN-γ, leading to a disbalance in immune surveillance mechanisms and changes in the tissue environment [38,39,40]. Candidalysin is a cytolytic toxin and was shown to induce NF-kB and MAPK pathways, which result most of the time as being dysregulated in cancer [41].

Sex and age distribution across our strata were similar to those figures reported in previous narrative reviews (i.e., predominantly men in their 50s and 60s) [42]. The buccal mucosa has been the most frequent location of CHC as classically reported, but other regions, such as the tongue or palate, may also be affected [1,3,4]. We could not report a preferential location in which squamous carcinoma may arise due to poor reporting information retrieved from the included studies. In terms of histopathology, and particularly regarding ED, only two studies reported its grade (Table 1 and Table 2) [6,22]. None of these studies confirmed that the presence of ED represented a risk factor for MT. On the other hand, several authors consider that *Candida* infection correlates with more pronounced grades of ED and thus indirectly implies a higher risk of malignancy [43]. Odell et al. considered how in CHC histopathology, it is important to identify epithelial changes as reactive or pre-neoplastic. This group also pointed out that re-biopsy should be delayed by 6 weeks after antifungal treatment bearing in mind the turnover time of the oral mucosal epithelium [11]. Following systemic antifungal treatment, CHC resolution has been reported with a reduction in ED [44].

Some potential limitations to our pooled analysis should be highlighted. Meaningful statistical between-study heterogeneity was found as expected, and priori planned random-effects estimations were used to be more conservative. This situation is extremely frequent in prevalent meta-analyses [45]. In this vein, heterogeneity presence should be considered more as the rule rather than the exception [46]. Subgroup analyses also showed that non-Asian and poor-quality studies tended to overestimate the MT rate of CHC. Subgroup analysis, according to RoB assessment, showed that results lacked robustness and probably underpowered our data synthesis in a moderate manner. Furthermore, it should be emphasized that only data from 271 individuals were included in this meta-analysis. In this vein, no conclusive results can be derived from these analyses of the context characteristics of CHC cancerization to avoid generalizations.

Secondly, publication bias (i.e., the tendency to publish only studies with positive results) may, in part, explain our results. Our methods lack statistical power due to the fact that the number of primary studies does not reach ten [47]. It is also worth mentioning that simulation analyses point out high type 1 error artifacts when assessing publication bias in prevalence meta-analyses and evaluating infrequent outcomes [48].

Thirdly, information bias should be considered. This concept refers to errors that are introduced during the exposure, event, or other co-variables assessment on each specific-study population [49]. The lack of consensus regarding CHC diagnostic criteria is likely to introduce measurement error and, therefore, misdiagnosis. In addition, in terms of the exposure to co-variables that may exert an effect on MT, heterogeneous reports were found regarding risk factors, including histopathology, which hindered more detailed analyses. This study may have a protopathic bias, which is detected when an exposure is influenced by early (subclinical) stages of a pathology [50]. Despite the explained mechanisms that could explain the carcinogenic potential of CHC, other explanations may be plausible such as the presence of a reverse causation. This is due to the lack of any threshold to stablish a differential diagnosis between CHC and a canonical OL without suffering from a fungal infection during its course. The lack of a clear differential diagnosis creates a two-way causal relationship implying a relevant feedback loop.

Finally, large heterogeneity can be caused by errors unrelated to the sample population. The range of sample types and techniques each has their limitations and may have contributed to this large heterogeneity. Each technique has a threshold of detection. Especially microscopy is more prone to overlook part of the sample as it is not entirely analyzed. It is well known that fungi can be well identified using microscopy without making a distinction on identity. Moreover, cultivation techniques may intentionally or unintentionally be targeted to a specific species, jeopardizing the possibility of identifying all fungi. To conclude, no medium contains enough nutrients to cultivate all fungi in a laboratory environment owing to the specific nutritional requirements and metabolic states of some species [51].

Despite the aforementioned limitations, the robust nature of the current pooled analysis is indicated by the forest plot, which demonstrates a consistent statistical malignant transformation rate for chronic hyperplastic candidiasis as its robustness is confirmed by sensitivity analysis.

## 5. Conclusions

The current available evidence on the malignant transformation rate of CHC remains limited. Nonetheless, the present systematic ascertained a pooled malignant transformation rate of 12.1% derived from a reduced number of longitudinal studies, and so avoiding the draw of broad inferences from these particular observations. CHC is broadly considered to be clinically important but a relatively neglected condition for the scientific community. Based on the current study’s estimates of malignant development, CHC-affected patients should be placed on exhaustive surveillance programs. In the author’s opinion, *Candida* infection should be seen as a relevant risk factor for OPMDs, particularly in OL. Nonetheless, it remains complex to affirm that CHC is an individual OPMD according to the evidence gathered in the present study. Prospective cohort studies to define the natural history of CHC and a consensus statement to clarify a proper set of diagnostic criteria are strongly needed.

## Figures and Tables

**Figure 1 jof-08-01093-f001:**
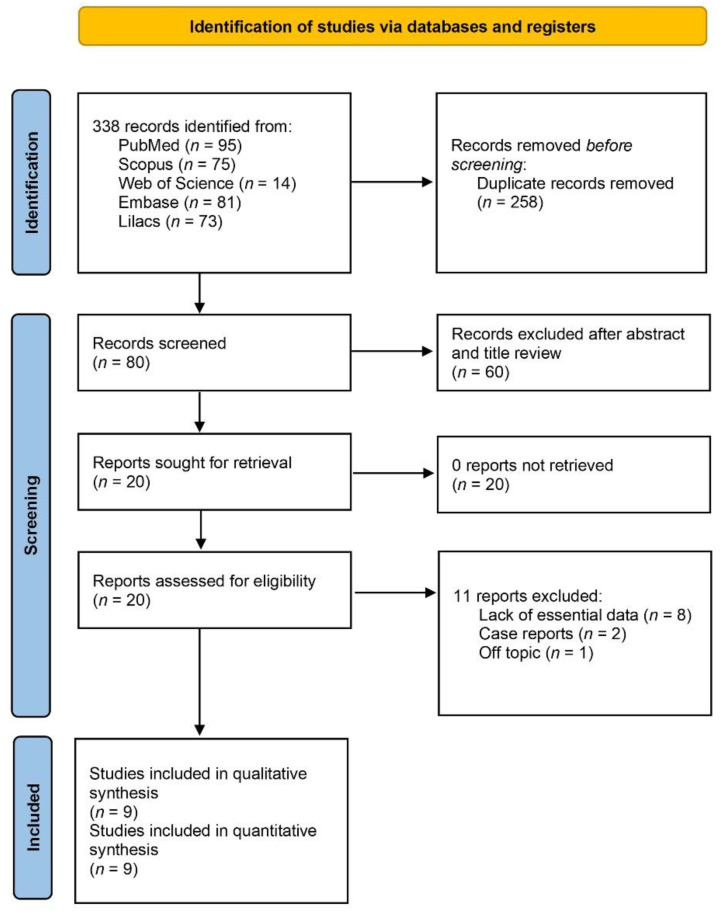
Prisma flow diagram of the searching processes.

**Figure 2 jof-08-01093-f002:**
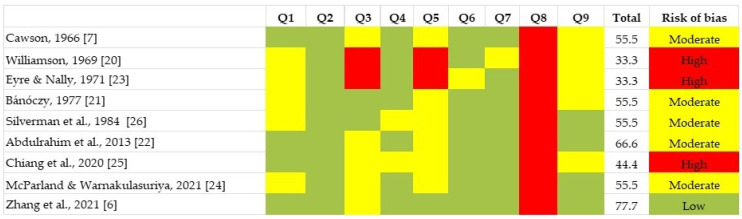
Quality plot graphically depicting the risk of bias among individual studies, assessed using Joanna Briggs Institute Critical Appraisal Checklist designed for systematic reviews addressing prevalence, cumulative incidence questions, and/or for proportion meta-analyses [6,7,20,21,22,23,24,25,26].

**Figure 3 jof-08-01093-f003:**
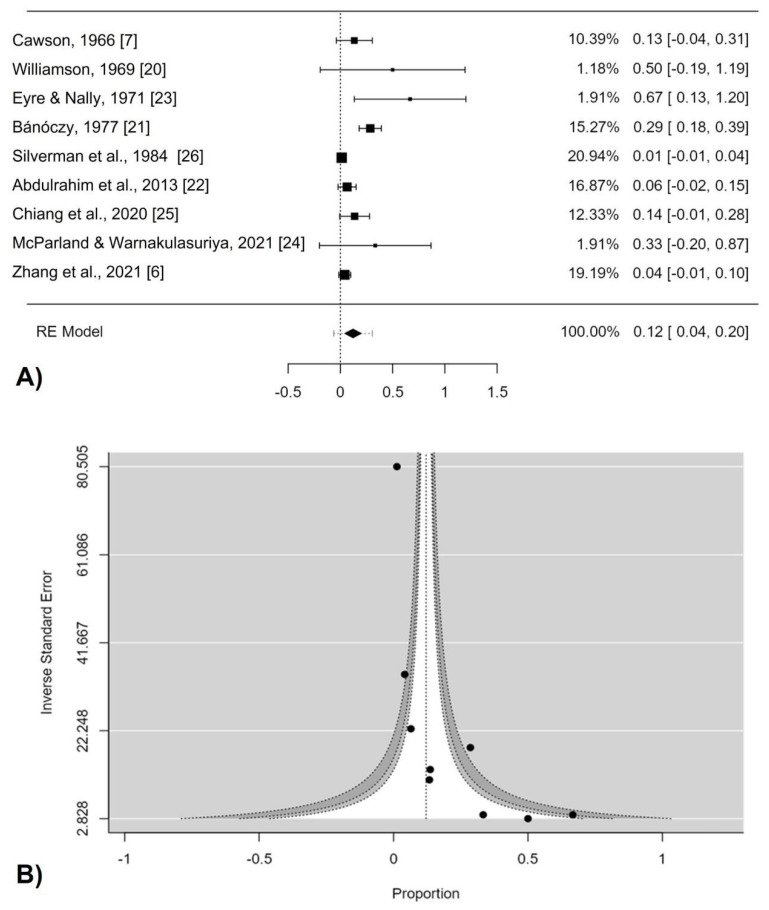
(**A**) Forest plot. Overall malignant transformation of chronic hyperplastic candidiasis. RE (random effects) and weight of each study; (**B**) Funnel plot assessing the publication bias [6,7,20,21,22,23,24,25,26].

**Table 1 jof-08-01093-t001:** Main clinicodemographic data of patients with CHC included in the study (F: female; M: male; *n*: number).

Author and Year	Country	Study Type	Patients				Follow-Up (Years)	Method of Assessment	Malignant Development (%)
			*n*	Sex		Mean Age (Years)			
				F	M				
Cawson, 1966 [7]	UK	Case series	15	4	11	50.5	10	Biopsy (P.A.S)	13.3
Williamson, 1969 [20]	UK	Case series	2	0	2	59.5	-	Biopsy (P.A.S), culture	50
Eyre & Nally, 1971 [23]	UK	Case report	3	0	3	55.7	-	Scraping and culture	66.6
Bánóczy, 1977 [21]	Hungary	Cohort	70	-	-	-	9.8	Biopsy (P.A.S)	28.7
Silverman et al., 1984 [26]	USA	Cohort	80	-	-	54	7.2	Biopsy (P.A.S), culture	4.4
Abdulrahim et al., 2013 [22]	Ireland	Case-control	31	14	17	57.8	2	Biopsy (P.A.S)	6.5
Chiang et al., 2020 [25]	Taiwan	Cohort	22	-	-	-	5	Biopsy (P.A.S),	13.6
McParland & Warnakulasuriya, 2021 [24]	UK	Descriptive	3	-	-	-	16	Biopsy (P.A.S),	33.3
Zhang et al., 2021 [6]	China	Cohort	48	13	35	54.9	6	Biopsy (P.A.S),	4.2
**Total**			274	31	68	55.4	8	-	12.4

**Table 2 jof-08-01093-t002:** Location of CHC lesions at the time of diagnosis (FOM: floor of the mouth; ED: epithelial dysplasia; CHC: chronic hyperplastic candidiasis; SCC: squamous cell carcinoma).

Author and Year	Location							Histopathology			
	FOM	Retromolar	Gingiva	Palate	Tongue	Buccal	Lip	SCC	High-ED	Low-ED	No ED
Cawson, 1966 [7]	0	0	0	4	4	6	1	-	-	-	-
Williamson, 1969 [20]	-	-	-	-	-	-	-	1	-	-	1
Eyre & Nally, 1971 [23]	-	-	-	-	-	2	3	2	-	-	-
Bánóczy, 1977 [21]	-	-	-	-	-	-	-	-	-	-	-
Silverman et al., 1984 [26]	-	-	-	-	-	-	-	2	-	-	-
Abdulrahim et al., 2013 [22]	0	0	2	3	4	22	-	2	8	18	3
Chiang et al., 2020 [25]	-	-	-	-	-	-		3	-	-	-
McParland & Warnakulasuriya, 2021 [24]	-	-	-	-	-	-	-	-	-	-	-
Zhang et al., 2021 [6]	-	-	-	-	-	-	-	0	1	9	38
**Total**	-	-	2	7	8	30	4	10	9	27	42

**Table 3 jof-08-01093-t003:** Pooled prevalence, malignant transformation rate, and subgroup analysis of the initial histopathological diagnosis of CHC (CI: confidence intervals; CHC: chronic hyperplastic candidiasis; PP: pooled proportion).

	Sample Size (*n*)		Pooled Data		Heterogeneity	
	Studies	Patients	PP (95% CI)	*p*-Value	*p_het_*	*I*^2^ (%)
**Malignant development**						
	9	274	PP = 12.1% (4.3–19.8)	0.002	0.001	78.27
**Subgroup analysis**						
Low and moderate risk of bias	6	247	PP = 9.8% (2.0–17.8)	0.014	0.001	81.98
High risk of bias	3	27	PP = 34.6% (−0.2–71.5)	0.06	0.113	54.21
Asian	2	70	PP = 6.5% (−1.5–14.5)	0.111	0.229	31.03
Non-Asian	7	204	PP = 16.3% (3.9–28.9)	0.010	0.001	82.77

**Table 4 jof-08-01093-t004:** Leave-one-out pooled prevalence analysis.

Studies	Pooled Proportion	95% CI
Overall	12.1	4.3–19.8
Omitting Cawson [7]	12.0	3.7–20.4
Omitting Williamson [20]	11.6	3.9–19.3
Omitting Eyre & Nally [23]	10.8	3.4–18.1
Omitting Bánóczy [21]	6.6	1.1–12.1
Omitting Silverman et al. [26]	15.6	5.9–25.3
Omitting Abdulrahim et al. [22]	13.9	4.6–23.1
Omitting Chiang et al. [25]	12.0	3.6–20.4
Omitting McParland & Warnakulasuriya [24]	11.7	3.9–19.5
Omitting Zhang et al. [6]	15.6	4.7–26.4

## Data Availability

The data that support the findings of this study are available from the corresponding author upon reasonable request.

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
