# Peer review of "Oral Chronic Hyperplastic Candidiasis and Its Potential Risk of Malignant Transformation: A Systematic Review and Prevalence Meta-Analysis"

_jof, 2022, doi:10.3390/jof8101093_

Round 1
Reviewer 1 Report
1) I would like to suggest that the treatment for candidiasis could be approached in this paper because 1 over 10 patients diagnosed of chronic hyperplastic candidiasis will eventually undergo a malignant transformation.
2) The authors could talk about briefly talk about candidiasis prevention in the text.
3) On line 288, the authors talk about the the e multifactorial context of oral carcinogenesis. As HIV + patients can develop oral candidiasis, is this population more susceptible to develop a malignant form?
Author Response
Reviewer: 1
First of all, we wish to thank the reviewer from his/her work and dedication to our article. His/Her useful comments and suggestions really contributed to the improvement of our manuscript.
1) I would like to suggest that the treatment for candidiasis could be approached in this paper because 1 over 10 patients diagnosed of chronic hyperplastic candidiasis will eventually undergo a malignant transformation.
Thank you for this consideration. We opted to include a new paragraph in the introduction addressing the current treatment modalities available for CHC, mainly dividing them in two medicament management and surgical methods.
2) The authors could talk about briefly talk about candidiasis prevention in the text.
We sincerely thank this appreciation. There are evidenced-based prevention methods for other kinds of oral candidosis. However, there is poor literature regarding prevention strategies in the case of CHC. We only have found that the elimination of the unhealthy habit of smoking of chewing tobacco, requesting patients not wearing their prosthesis at night and improve oral hygiene. Also, some authors pointed out that anemia and avitaminosis might also contribute to CHC onset. As requested, we now briefly address this point.
3) On line 288, the authors talk about the multifactorial context of oral carcinogenesis. As HIV + patients can develop oral candidiasis, is this population more susceptible to develop a malignant form?
We sincerely appreciate this question. It is well established that HIV+ patients harbor increased levels of Candida colonizing the oral cavity and are significantly predisposed suffer from some oral candidiasis subtypes. Oral candidosis is also related to other types of clinical situations that weaken the immune system such as immunosuppressive therapies, diabetes, T-cell deficiency, or severe combined immunodeficiency, DiGeorge syndrome, hereditary myeloperoxidase deficiency or Chediak-Higashi syndrome. We have carefully reviewed all the articles included in the present systematic review as other narrative reviews included as references in the present manuscript and we could not find any patient affected simultaneously by a HIV infection and CHC. In this vein, we cannot ask accurately to the raised question. In addition, we have ascertained that HIV+ patients are more prone to other forms of oral candidiasis such as pseudomembranous and erythematous types.
All the modifications made in order to address your comments and suggestions were indicated with a yellow background.
Once again, we would like to express our gratitude for your constructive criticism.
Sincerely
Reviewer 2 Report
Lorenzo-Pouso et al. performed a systematic review and meta-analysis to evaluate the potential risk of malignant transformation of the Chronic hyperplastic candidiasis (CHC), a subtype of oral candidiasis. Based on this analysis, the authors estimated the malignant transformation rate of CHC and related factors. The article brings important information, and the text is well written. Only few parts need revisions:
-Line 19: The objective (“We aimed to estimate the MT rate of CHC”) must be rewritten with more details.
-Lines 24-27: This sentence is unclear: “The overall MT rate for CHC across all studies was 12.1% (95% confidential interval, 4.1%–19.8%) …… MT pooled proportion was 12.1%”.
-Line 29-30: Conclusion in the abstract needs to be more specific. I think that the conclusion of abstract can follow the idea of the last phrase of the paper: “Prospective cohort studies to define the natural history of CHC and a consensus statement to clarify a proper set of diagnostic criteria are strongly needed.
-Line 62: Describe OL.
-Lines 72-75: This part needs improvement. The phrases should be better connected.
-Line 96: Correct the word “e-amil”
-Lines 250-252: These two phrases are unclear: “Egger's regression test confirmed more formally the existence of ¨small study effects¨ (pEgger = 0.001). In the subgroup analysis.”
-Lines 351-353: Clarify the sentence: “Although, because our strata contain heterogeneous studies with low event rates, also some of them were not initially intended to assess the outcome under study”.
-In the discussion section, add comments about the low number of studies that performed cultures and/or species identification.
-In general, the name of microorganisms must be corrected and written in italic.
Author Response
First of all, we wish to thank the reviewer from his/her work and dedication to our article. His/Her useful comments and suggestions really contributed to the improvement of our manuscript.
Lorenzo-Pouso et al. performed a systematic review and meta-analysis to evaluate the potential risk of malignant transformation of the Chronic hyperplastic candidiasis (CHC), a subtype of oral candidiasis. Based on this analysis, the authors estimated the malignant transformation rate of CHC and related factors. The article brings important information, and the text is well written. Only few parts need revisions:
-Line 19: The objective (“We aimed to estimate the MT rate of CHC”) must be rewritten with more details.
We sincerely thank this suggestion. As requested, the objective was rewritten to ensure clarity.
-Lines 24-27: This sentence is unclear: “The overall MT rate for CHC across all studies was 12.1% (95% confidential interval, 4.1%–19.8%) …… MT pooled proportion was 12.1%”.
Thank you for appreciating this mistake. This has been an unwitting error on our side and as requested it was amended.
-Line 29-30: Conclusion in the abstract needs to be more specific. I think that the conclusion of abstract can follow the idea of the last phrase of the paper: “Prospective cohort studies to define the natural history of CHC and a consensus statement to clarify a proper set of diagnostic criteria are strongly needed.
Thank you for this consideration. We also believe that the way of expressing a conclusion at the end of the manuscript is clearer than that used in the abstract. In this vein, it was changes as requested.
-Line 62: Describe OL.
Thank you for this suggestion. We have briefly described OL and some more related information.
-Lines 72-75: This part needs improvement. The phrases should be better connected.
Thank you for this suggestion. We have tried to be more clear in our objectives description.
-Line 96: Correct the word “e-amil”
Amended.
-Lines 250-252: These two phrases are unclear: “Egger's regression test confirmed more formally the existence of ¨small study effects¨ (pEgger = 0.001). In the subgroup analysis.”
Thank you for appreciating this mistake. We have amended it as requested.
-Lines 351-353: Clarify the sentence: “Although, because our strata contain heterogeneous studies with low event rates, also some of them were not initially intended to assess the outcome under study”.
We believe that this sentence is unfortunate in the general context of the manuscript and we have chosen to delete it.
-In the discussion section, add comments about the low number of studies that performed cultures and/or species identification.
As requested he have addressed this point as limitation at the end of our discussion.
-In general, the name of microorganisms must be corrected and written in italic.
Amended.
All the modifications made in order to address your comments and suggestions were indicated with a blue background.
Once again, we would like to express our gratitude for your constructive criticism.
Sincerely
Author Response
The authors completed an online registry of meta-analyses, which were written in accordance with PRISMA standards and in accordance with the criteria for writing systematic reviews. This article discusses the risk of malignant transformation (MT) of oral chronic proliferative candidiasis (CHC), which has demonstrated the CHC may serve as the remarkable risk factor of MT. This finding is of clinical significance both for the CHC patients prognosis and oral carcinogenesis. My suggestions are as the following:
Major suggestion:
- It seems the present study did not through the Cochrane library, which serves as the gold standard of meta-analyses;
Cochrane library database was not included in the original search because collecting clinical trials. This study is based on prevalence retrospective studies. However, we performed a search and only one result about CHC popped up, which did not meet inclusion criteria. https://www.cochranelibrary.com/advanced-search by input “Chronic hyperplastic candidosis”. https://www.cochranelibrary.com/central/doi/10.1002/central/CN-00732052/full
- The quantity and quality of the included literature were relatively low, and the proportion of “high-risk” literatures (which may influence the quality of the analyses)was high, however, the authors already stated this point in the discussion part
Thanks for your precious comment. This will be useful for future studies and standardization in future research.
- The heterogeneity was high, there was a large degree of bias, and the subgroup analysis was not sufficient. For example, if smoking plays an important role in the development of oral malignancies? Which fungal pathogen is more related to the MT? To answer these question, subgroup analysis seems to be warranted;
Oral carcinogenesis is a complex and multi-factorial process. It’s well known that smoking can contribute to oral carcinogenesis, but not all the included studies reported information about smoking status of patients, so that subgroup analysis could not be performed. Primary literature, only reported smoking among all their samples not individually in patients with MT or without MT. If that was the case, we could have performed a metaregression as we did with the year of publication or tried a subgroup analysis as you suggested but with this initial literature is not impossible even to get to a qualitative evidenced-based recommendation.
Moreover, current studies only investigate Candida albicans, while other species are still under-investigated. Only a single study denoted two other species C glabrata and C tropicalis, but the case affected by these types did not undergo a MT (Zhang et al. 2021). In this vein, lamentably these data are impossible to pool. This issue has been added as a limitation according to previews reviewer 2 suggestion.
- In addition to the clinical research results analysis, the discussion can be extended to explore the possible pathogenesis of the disease.
We are really thankful for this suggestion. We added pathogenesis into the discussion as follow: Indeed, nowadays different theories exist on candida infection mechanisms linked to oral carcinogenesis. Candida can produce nitrosamines, such as N-nitroso-benzyl-methylamine and acetaldehyde, which are cancerogenic com-pounds . Moreover, during infection there is an increase of proinflammatory cytokines (interleukin (IL)-1α, IL-1β, IL-6, IL-8, IL-18, tumor necrosis factor (TNF)-α, IFN-γ, leading to disbalance in immune surveillance mechanisms and changes in the tissue environment . Candidalysin is a cytolytic toxin and was shown to induce NF-kB and MAPK pathways, pathways resulting most of the times dysregulated in cancer.
Minor suggestion:
- Tables 3 and 4 should be in standard form.
We sincerely thank this suggestion. All tables are provided as modifiable items embedded in the manuscript. MDPI as editorial takes care of its changes in case of acceptance or will be in touch during proofreading.
- The first letter of Candida should be always in capital form.
Amended.
- The Latin name such as Candida, Candida albicans etc. should be in italic.
Changed according your suggestion
All the modifications made in order to address your comments and suggestions were indicated with a green background.
Once again, we would like to express our gratitude for your constructive criticism.
Sincerely